# Assessing the reliability of a digital inclinometer app for measuring hip, knee, and ankle proprioception

Sophia G. Chirumbole[1], Rachel H. Teater[2], Megan M. Bals[3], Daniel R. Richie[4], Scott M. Monfort[5], Ajit M. W. Chaudhari[1,4,6]*

1 Mechanical & Aerospace Engineering, Ohio State University, Columbus, Ohio, United States of America,
2 Mechanical Engineering, Vanderbilt University, Nashville, Tennessee, United States of America,
3 Physical Therapy, NovaCare Rehabilitation, United States of America, 4 Biomedical Engineering, Ohio State University, Columbus, Ohio, United States of America, 5 Mechanical & Industrial Engineering, Montana State University, Bozeman, Montana, United States of America, 6 Health and Rehabilitation Sciences, Ohio State University, Columbus, Ohio, United States of America

* Ajit.Chaudhari@osumc.edu

**Data Availability Statement:** All relevant data are within the manuscript and its Supporting information files.

## Abstract

Proprioception can be defined as the ability of an individual to detect motion and position of the various joints in their bodies. Current tools for measuring proprioception lack consensus on their accuracy and validity; they also each have their own limitations, and, furthermore, present barriers to use for clinicians. We propose a new and reliable method for evaluating hip, knee, and ankle proprioception by utilizing a digital inclinometer app to measure joint position sense. The digital inclinometer app recorded the active joint position sense error after each of five trials for the hip and knee joint and ten trials for the ankle joint. To quantify the reliability of the digital inclinometer app, single-measurement and average-measurement intra-class correlation coefficients (ICC) along with the associated 95% confidence intervals (95% CI) were calculated for each joint's position sense error across trials. Both the hip (ICC (2,k) = 0.849 (95% CI = [0.783–0.897])) and knee joint (ICC (2,k) = 0.837 (95% CI = [0.750–0.897])) were found to have moderate to good reliability when the middle three of five trials were analyzed. Unlike the hip and knee, moderate to good reliability for ankle proprioception (ICC (2,k) = 0.785 (95% CI = [0.539–0.893])) was only achieved with the middle eight of ten trials. The results of this study indicate that this digital inclinometer app is able to accurately record joint position sense at the hip, knee, and ankle when the appropriate number of trials are collected; thus, allowing this tool and methodology to be considered for use in both clinical and research environments to measure proprioception, and furthermore, quantify proprioceptive deficits.

## Introduction

Proprioception involves the awareness of joint position and movement as well as the integration with other somatosensory, visual, and vestibular information to coordinate the individual's movement patterns [1]. Although there is no consensus among researchers to define

**Funding:** The author(s) received no specific funding for this work.

**Competing interests:** The authors have declared that no competing interests exist.

proprioception, it can be generally described, as an "individual's ability to integrate the sensory signals from mechanoreceptors to thereby determine body segment positions and movements in space" [1]. One key function of human movement that depends on proprioception is postural control, as those with proprioceptive deficits often have increased postural sway and decreased single leg balance [2–4]. Some researchers suggest deficits in proprioception correlate with increased risk for injury, decreased muscle strength and power, and long-term musculoskeletal disorders, such as osteoarthritis [2]. Although proprioception is presumed to be important to posture and movement, the literature in this area is limited, which is in part due to the lack of accepted standard methods for measuring it.

There are two common methods of measuring proprioception: threshold to detect passive motion (TDPM) and joint position sense (JPS) [5]. TDPM is measured by a tester passively moving a participant's joint in a direction (e.g., flexion, extension, etc.) and the participant then indicating when they sense movement of the joint. During JPS testing, the participant must remember and reproduce a target angle. There are two types of JPS testing, active and passive. Active joint position sense (aJPS) testing requires participants to remember a joint position angle and then attempt on their own to reproduce the joint angle. In passive joint position sense (pJPS) testing, the participant is still required to remember an angle, but the tester then moves the joint through the range of motion until the participant states they are at the remembered angle [6]. Unfortunately, TDPM and JPS methods of measuring proprioception are not directly comparable [7,8]. TDPM has been previously shown to be a more reliable measurement of proprioception, but it has less ecological validity compared to aJPS, as it depends solely on processing sensory feedback [1,8]. The aJPS method may be more functional and indicative of true proprioceptive accuracy [7,9] because it requires both sensorimotor integration and motor control [8]. In the current study, we chose to look at aJPS because functional proprioception during daily tasks always includes both sensorimotor and motor control components.

Evaluation of proprioception is important for rehabilitation, causing both clinicians and researchers alike to call for a new protocol to measure proprioception [7]. Proprioception is an increasingly important component of rehabilitation and has been shown to improve with targeted exercises [10]. Awareness of proprioceptive deficits is beneficial to clinicians to help focus interventions specific to patients' impairments and allow for optimal recovery. However, a lack of data exists about the accuracy and validity of available tools [8,11]. Isokinetic dynamometers, such as the Biodex system, are commonly assumed to be a gold standard for many assessments of function including strength and proprioception testing [1]; however, this equipment is expensive, bulky, and not portable. Additionally, the straps, pads, and the inertia of the Biodex arm may provide additional somatosensory cues to the participant, which potentially make it a less ecologically valid assessment of joint proprioception. Electro-goniometers, potentiometers, inclinometers, and 2D and 3D motion analysis have all also been shown to take reliable joint angle measurements to determine proprioception [12–14]; however, these measurement tools require extensive data analysis and expensive equipment for interpretation [11]. These limitations can present clinicians with a barrier to measuring joint proprioception in populations that are at risk for having deficits. Recently, mobile apps have increasingly been investigated as means to quantify human movement for activity monitoring, functional assessment, and rehabilitation purposes [15]. However, the reliability and validity of these apps for assessing proprioception are currently unknown.

In the current article, three separate datasets were collected to establish the within-session reliability of app-based digital inclinometry for measuring aJPS. In addition to establishing within-session reliability, each joint dataset was analyzed separately for an additional secondary interest, or purpose. The secondary interest in performing the hip proprioception testing

was to determine if conclusions can be drawn for both limbs from single limb hip aJPS error data (e.g. if one can gain information about the right limb, even if aJPS data is only collected on the left limb). Previously, the knee joint active repositioning angle error has been analyzed for between limb differences, but less so for the hip joint [16]. The secondary interest in performing the knee proprioception testing was to determine if knee aJPS error is consistent over time, more specifically, when trials are separated by an intervening activity, or a delay in time. This has been done previously by comparing data collected on separate days with the gleno-humeral joint [17], and understanding the inter-session reliability of aJPS measurements is critical to use in clinical settings to track longitudinal changes. Lastly, the secondary interest in performing the ankle proprioception testing was to determine how many trials were needed to achieve acceptable reliability. In current literature, there exists no gold standard for the appropriate number of trials to collect to achieve acceptable reliability of ankle JPS using a digital inclinometer app [18].

This article reports the reliability and validity of a digital inclinometer app that has been previously validated for quantifying pelvic motion. When compared with motion capture, considered a "gold standard" in this previous study, the digital inclinometer app was found to have good reliability for measurement of pelvic motion [19]. In the current article, additionally, inter-limb and inter-session differences are explored in the hip and knee joints. Establishing the reliability and validity of app-based digital inclinometry for quantifying joint proprioception, including any inter-limb and inter-session significance, will enable researchers and clinicians to easily and cost effectively assess joint proprioception in individuals who may have proprioceptive deficits for a variety of patient populations.

## Methods

### Study participants

This article describes the analyses of three separate datasets collected to understand hip, knee, and ankle aJPS reliability. Deficits in memory can impede a participant's ability to recall the position independent of proprioceptive deficits and can confound data for this measurement [1]; therefore, only participants without any known cognitive impairment were included.

**Hip proprioception.** A convenience sample of forty-two participants were recruited and tested between the months of March and April 2017 at the Ohio State University. A total of 17 males and 25 females were included in this portion of the study, with ages ranging from 21–31 years old. Hip proprioception data were originally collected as part of an educational assignment for a course, so neither written or verbal informed consent was not obtained for these participants. Inclusion criteria included enrollment in a course on research methods for first-year Doctorate of Physical Therapy students, ability to fully bear weight on both lower extremities, no known cognitive impairments, and no known prior hip pathology. The Ohio State University Biomedical IRB approved subsequent use of these data for research (2023H0145).

**Knee proprioception.** Thirty healthy participants were recruited and tested between the months of August and December 2018 at the Ohio State University. This sample size is larger than the projected minimum of twenty-six participants, that provides 0.8 power to detect an ICC (2,k) = 0.8 compared to a null hypothesis of ICC (2,k) = 0.6 that will be used to demonstrate the test-retest reliability of the iPod app assessment. A total of 19 males and 11 females were included in this portion of the study, with ages ranging from 22–27 years old. Inclusion criteria included no history of knee pain, no history of nerve damage in the lower extremities, no known cognitive impairments, and no previous knee surgery. All participants provided Ohio State University Biomedical IRB approved written informed consent before participating (2017H0370).

**Ankle proprioception.** Forty-six healthy participants were recruited and tested in August of 2017 at the American Society of Biomechanics Annual Meeting in Boulder, Colorado. This was the maximum number of participants able to have their data recorded within the time period allotted for data collection during the conference. Demographic information, such as age and biological sex, was not recorded. Inclusion criteria included no known cognitive impairments and no history of nerve damage in the lower extremities. All participants provided Ohio State University Biomedical IRB approved written informed consent before participating (2017H0149).

## Instrumentation

This study used an iOS digital inclinometer app that measures an iPod's orientation (i.e., allowing us to measure the angle of one body segment relative to an earth-centered vertical or horizontal axis). All iPod models starting in 2011 have built-in triaxial accelerometers and gyroscopes along with software-based sensor fusion that allow them to estimate the orientation of the device as it moves in space [20]. Five trials at the hip and knee joints and ten trials at the ankle joint were recorded to determine the digital inclinometer app's reliability. The CoreX Therapy app was used because it allows zeroing of the inclination at the starting position. The app reports both the largest deviation and the final deviation from starting position when recording is stopped and allows the use of the Play/Pause button on a wired or Bluetooth headphone or speaker to start and stop the measurement. This app also permitted unlimited storage of trials within the app for later recall and analysis. The CoreX Therapy app used Apple's built-in CoreMotion sensor fusion to estimate device orientation with no additional low-pass filtering of the orientation angle. [21] Participants were unable to view the screen of the iPod during testing so they could not use the app for any cues as to their position or position error.

## Procedure

**Hip proprioception.** For all participants, a coin flip was used to randomize which limb would be tested first. The participants were blindfolded to remove visual information and allow for more accurate proprioception measurement, and headphones were placed in their ears to prevent them from hearing. Next, a belt holding an iPod with the digital inclinometer app installed was attached to the leg being tested, halfway between the anterior superior iliac spine and the patella (Fig 1). In one hand the participants held the button on the cord of their given pair of headphones, which were connected to the iPod running the app, and were instructed on how to use the headphone button to stop the app and record their data. Each participant was led through the experimental protocol (Fig 1). This process was repeated for a total of 5 trials on each leg.

**Knee proprioception.** For all participants, a coin flip was used to randomize which knee would be tested. Once the test limb had been established, it remained the test limb for the duration of the study. Participants were instructed to wear shorts to minimize external sensation from clothing.

For knee proprioception measurements, the iPod was secured to the participant's mid-tibia with a felt strap. A Bluetooth speaker connected to the iPod was handed to the participant so that they could reach the "play/pause" button, which was used to operate the test in the app. Each participant was asked to stand on a short box placed directly in front of a wall with their non-tested leg to allow the tested foot to clear the floor. The tested extremity's anterior thigh was pressed against the vertical wall in front of them (to achieve vertical orientation of the thigh) with the knee fully extended (Fig 2). The participant wore a blindfold to remove visual information and allow for more accurate proprioception measurement.

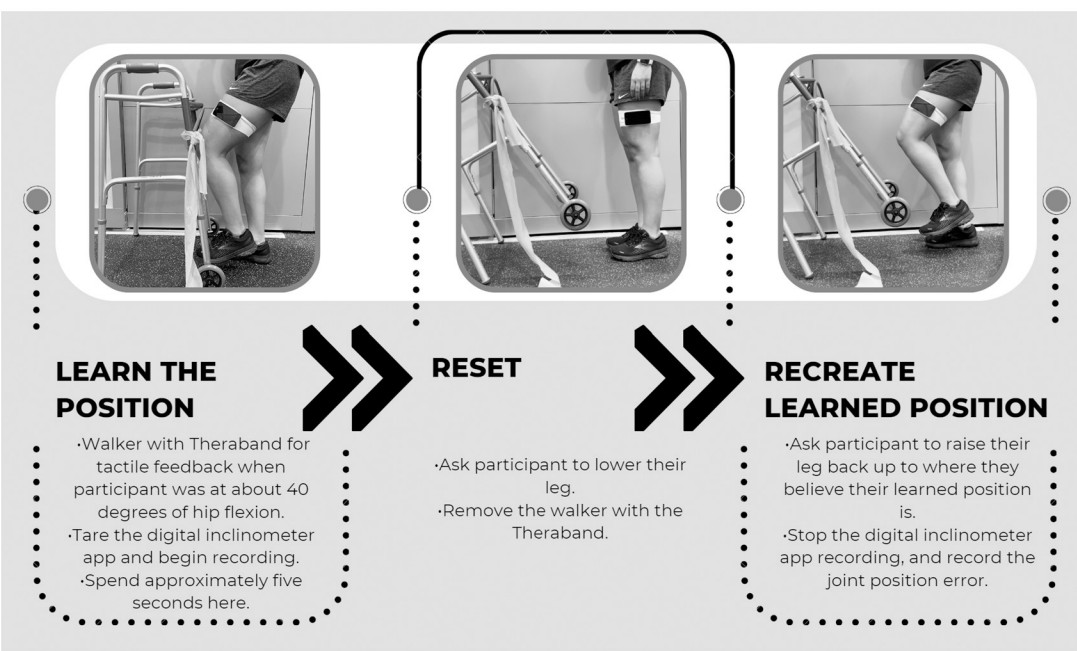

**Fig 1. Hip proprioception experimental protocol.**

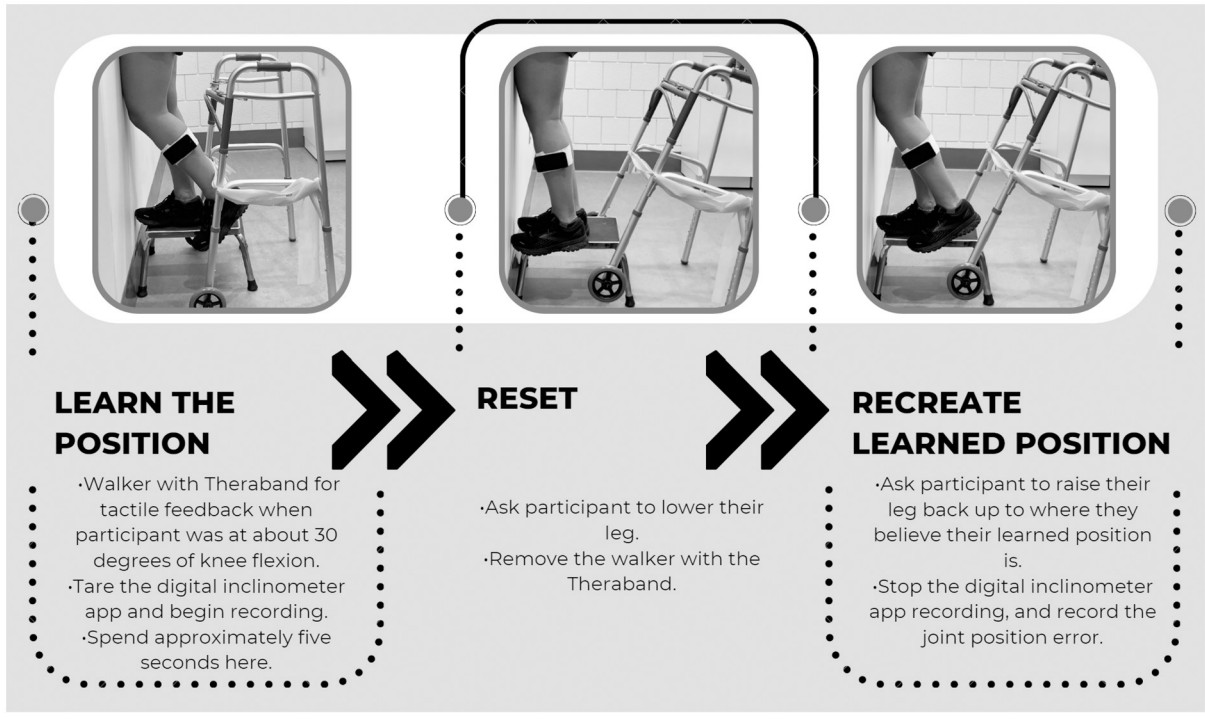

**Fig 2. Knee proprioception experimental protocol.**

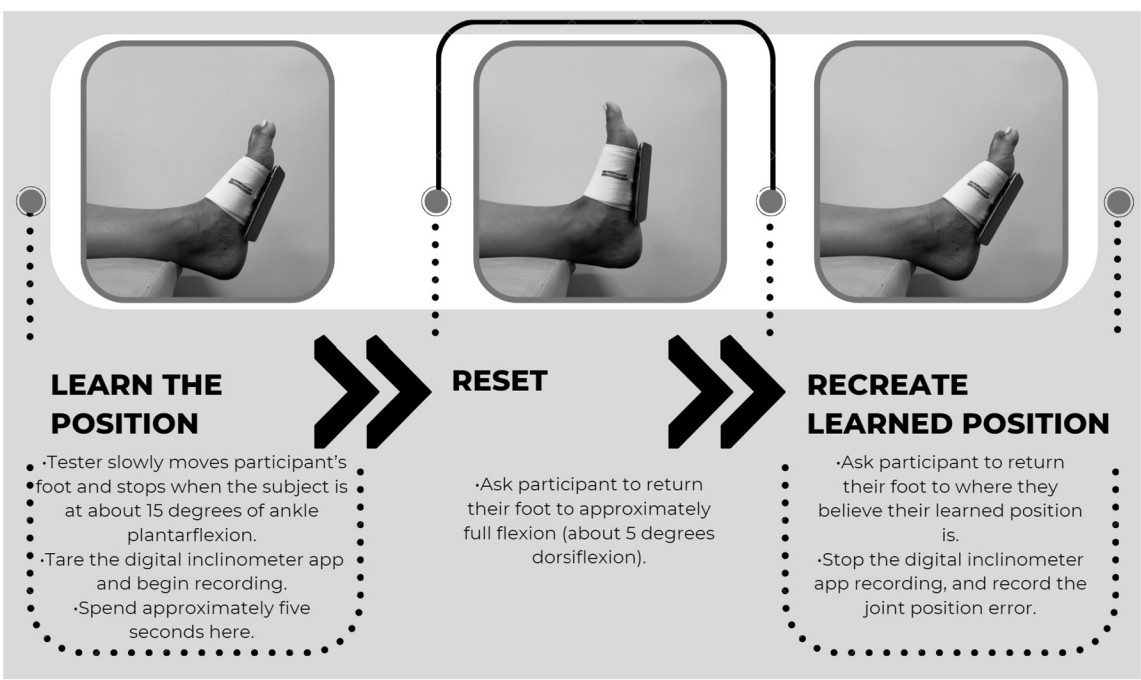

**Fig 3. Ankle proprioception experimental protocol.**

Before initiating the first trial, the participant was taken through a step-by-step walkthrough of the test to ensure understanding of the protocol. After the walkthrough, each participant went through five trials of active range of motion within the experimental protocol (Fig 2). These five trials were recorded. Each participant then performed other tasks, including single-leg landings and isokinetic strength testing, for a few minutes before returning to repeat the entire protocol listed above for a second time (later referred to as delay).

**Ankle proprioception.** Participants were asked to remove the shoe from the foot being tested which was randomly chosen by a coin flip. Participants were instructed to wear socks and roll up their pants, if they cover their ankles, so that the ankle was clear and observable. Participants were instructed to lie supine on a table or treatment plinth. For ankle proprioception measurement, the iPod was secured to the participant's foot with a felt strap (Fig 3). iPod headphones were plugged into the iPod and handed to the participant so that they could reach the "play/pause" button, which was used to operate the test in the app.

Before initiating the first trial, the participant was taken through a step-by-step walkthrough of the test to ensure understanding of the protocol. After the walkthrough, the participant went through a trial of active range of motion, following the experimental protocol (Fig 3). The participant's eyes were closed throughout the trials. This protocol was repeated nine more times, for a total of ten trials per participant.

## Data and statistical analysis

For each joint, the aJPS error is reported as an absolute value, in degrees, from the start position. The intraclass correlation coefficient (ICC) and associated 95% confidence interval (95% CI) were calculated between aJPS error values because they are commonly used to assess consistency across, or within, a dataset. [22] The ICC numerical values themselves should be identical whether one uses ICC (2,k) or ICC (3,k). The importance of the distinction between the

two lies in the interpretation of the data. ICC (2,k) assumes random-effects, which makes its interpretation generalizable across populations, whereas ICC (3,k) assumes fixed-effects. [23] Therefore, ICC (2,k) is reported here to determine if the aJPS measurements are generalizable across a greater participant population, as well as among other raters, or testers. Additionally, ICC (2,1) is reported to determine the data's validity and reliability if only a single measure is recorded. This is applicable in the current article, as it is suggested that this assessment tool may be used in clinical settings, where efficiency is important.

The ICC (2,1) and (2,k), as well as the 95% CI, reliability statistics were calculated for all recorded trials using SPSS Statistics software with a two-way random model and absolute agreement type (SPSS Statistics for Windows, Version 28.0. Armonk, NY: IBM Corp). The ICC (2,k) reliability statistic was calculated for the remaining data points after removing the minimum and maximum value for each testing session. The ICC (2,k) test applied to the remaining data points was used as the primary measure of test-retest reliability of the digital inclinometer app. Reliability was determined based on Portney & Watkins' recommendation that ICC values less than 0.5 are indicative of poor reliability, values between 0.5 and 0.75 indicate moderate reliability, values between 0.75 and 0.9 indicate good reliability, and values greater than 0.90 indicate excellent reliability [24]. Additionally, the standard error of measurement (SEM) was calculated for the aJPS error values at each joint. The SEM was calculated by taking the square root of the residual mean square error value, given by the SPSS software, for each separate reliability analysis, accordingly [25].

To test our secondary hypotheses for each joint, further analyses were performed for both the hip and knee joints. In the hip joint, to explore consistency across limbs, the ICC (2,1) statistic was calculated for the "trimmed average" after removing the minimum and maximum aJPS error value for the testing session done on each limb. This ICC (2,1) test applied to the trimmed average was used as the measure of aJPS error consistency over the left and right limbs at the hip joint. As for the knee joint, to explore any differences over time, the ICC (2,1) statistic was calculated for the "trimmed average" after removing the minimum and maximum value for the testing session done before and after a delay in time, or intervening activity. This ICC (2,1) test applied to the trimmed average was used as the measure of aJPS error consistency over time at the knee joint.

## Results

### Hip proprioception

The hip aJPS error ranged between 0.1˚ and 10.6˚ from the learned position. The average error from the starting position was 2.4˚ with a standard deviation of 1.7˚. Taking all five trials into account, the ICC (2,1) between trials was 0.388 (95% CI = [0.288–0.499]). After elimination of the largest two outliers within each participant, the ICC (2,k) was 0.849 (95% CI = [0.669–0.833]) (Table 1). The ICC (2,1) value represents poor reliability; however, the ICC (2,k) value represents good reliability. The scatter plot illustrates that with increasing aJPS error at the hip, variance increases (Fig 4). The inter-limb ICC (2,1) was found to be 0.508 (95% CI = [0.245–0.701]), which shows moderate agreement between limbs (Table 1).

### Knee proprioception

The knee aJPS error ranged from 0˚ to 8.5˚, with the average and standard deviation in error being 1.8˚ and 1.4˚, respectively. Taking all five trials into account the ICC (2,1) between trials was 0.276 (95% CI = [0.165–0.410]) (Table 1). After elimination of the largest and smallest error values within each participant, the ICC (2,k) of the middle 3 of 5 trials was 0.837 (95% CI = [0.750–0.897]) (Table 1). Similarly to the hip proprioception data, the ICC (2,1) of the knee

**Table 1. ICC, 95% CI, and Standard Error values demonstrating the reliability of the proprioception measurement for each joint location.**

| Joint | Number of Trials | ICC Test | ICC Value | 95% Confidence Interval | Standard Error (deg.) |
|---|---|---|---|---|---|
| Hip | 5 trials | ICC (2,1) | 0.388 | [0.288–0.499] | 1.6 |
| | | ICC (2,k) | 0.760 | [0.669–0.833] | |
| Hip | Middle 3 of 5 trials | ICC (2,k) | 0.849 | [0.783–0.897] | 1.0 |
| Hip | Inter-limb "trimmed average" | ICC (2,1) | 0.508 | [0.245–0.701] | 1.1 |
| Knee | 5 trials | ICC (2,1) | 0.276 | [0.165–0.410] | 1.2 |
| | | ICC (2,k) | 0.656 | [0.497–0.777] | |
| Knee | Middle 3 of 5 trials | ICC (2,k) | 0.837 | [0.750–0.897] | 0.7 |
| Knee | Pre & post time delay "trimmed average" | ICC (2,1) | 0.429 | [0.083–0.682] | 0.7 |
| Ankle | 10 trials | ICC (2,1) | 0.176 | [0.099–0.289] | 2.3 |
| | | ICC (2,k) | 0.682 | [0.524–0.803] | |
| Ankle | First 5 trials | ICC (2,k) | 0.481 | [0.200–0.685] | 2.3 |
| Ankle | Middle 3 of first 5 trials | ICC (2,k) | 0.585 | [0.324–0.757] | 1.4 |
| Ankle | Middle 8 of 10 trials | ICC (2,k) | 0.785 | [0.539–0.893] | 0.9 |

represents poor reliability; however, the ICC (2,k) of the knee represents moderate to good reliability. The scatter plot illustrates that with increasing aJPS error at the knee, variance increases (Fig 5). When comparing between trials before and after the delay, the ICC (2,1) was 0.429 (95% CI = [0.083–0.682]) (Table 1). This value indicates poor repeatability between sessions.

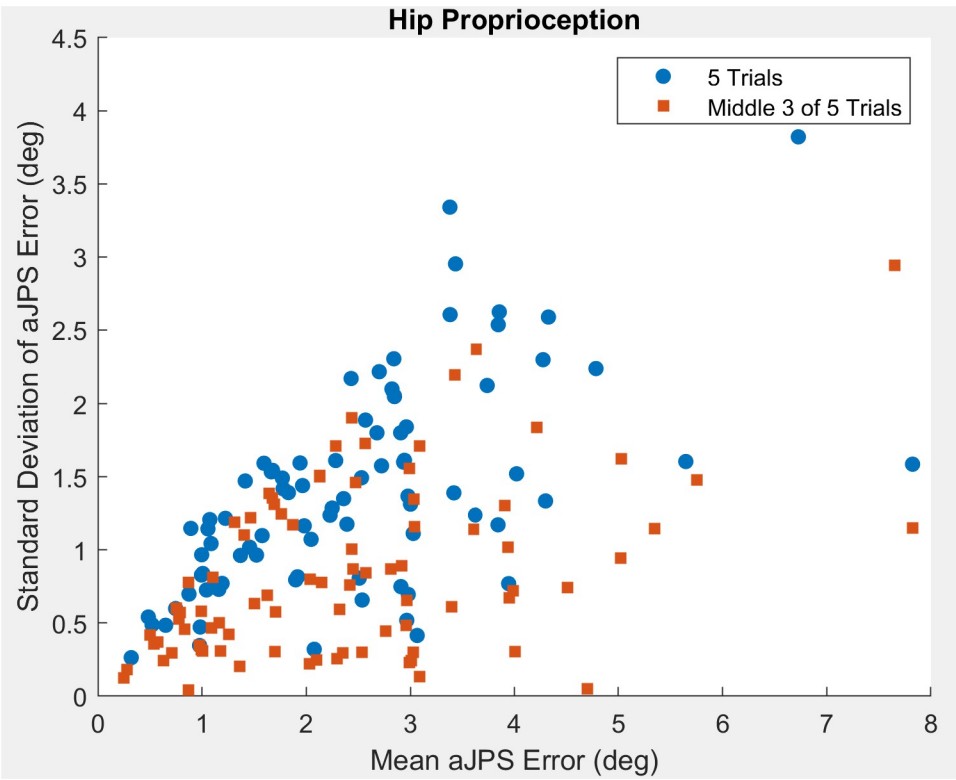

**Fig 4. Scatter plot comparing hip aJPS error standard deviation and mean aJPS values measured by the digital inclinometer application.**

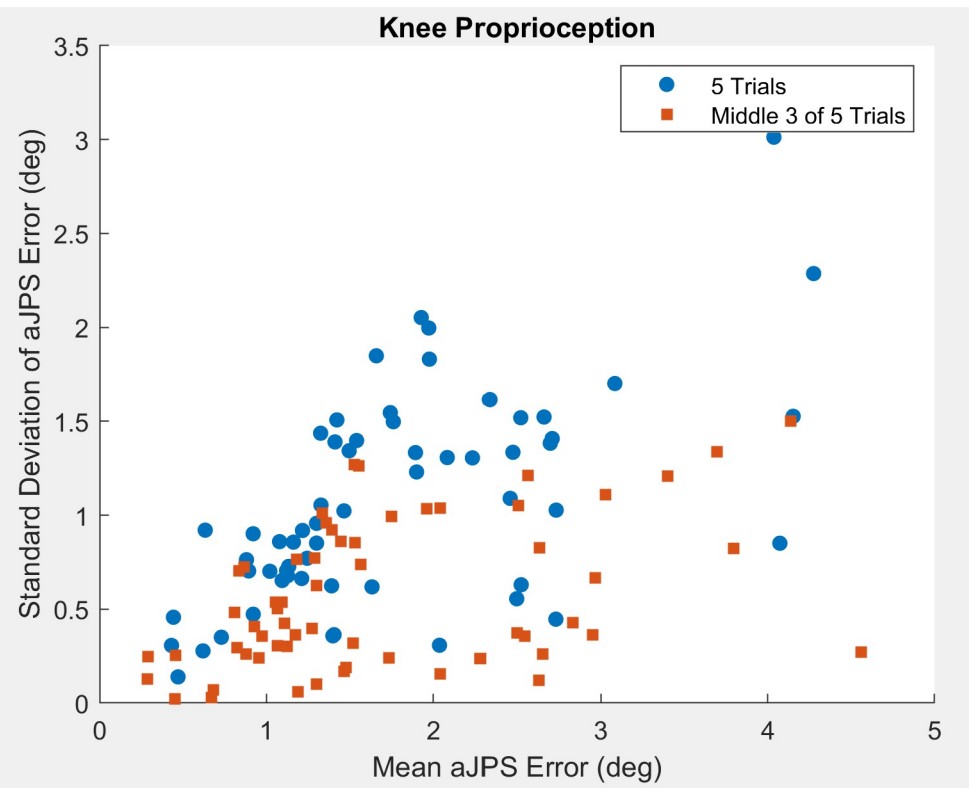

**Fig 5. Scatter plot comparing knee aJPS error standard deviation and mean aJPS values measured by the digital inclinometer application.**

## Ankle proprioception

The ankle aJPS error ranged from 0˚ to 18.6˚, with the average and standard deviation in error being 2.9˚ and 1.3˚, respectively. Including all 10 trials, the ICC (2,1) was 0.176 (95% CI = [0.099–0.289]), indicating poor reliability for a single-measurement approach. After elimination of the highest and lowest value out of the first 5 trials collected for each participant, the ICC (2,k) was 0.585 (95% CI = [0.324–0.757]) (Table 1). This indicated that, unlike the hip and knee proprioception data, with only 5 trials the ICC (2,k) of the ankle demonstrated poor to moderate reliability. Increasing the number of trials to the middle 8 of 10 trials, the ICC (2,k) increased to 0.785 (95% CI = [0.539–0.893]), indicating moderate to good reliability (Table 1). The scatter plot shows that with increasing aJPS error at the ankle, variance increases (Fig 6).

Additionally, directional systematic bias during trials was concluded to be negligibly low, illustrated by a sample Bland-Altman plot and histogram (Figs 7 & 8). The histogram below shows that the distribution of the raw ankle aJPS error data points are centered around zero on the x-axis (Fig 8). This allows the conclusion to be drawn that there exists negligible bias of over- or under-estimation of ankle aJPS during the trials. Furthermore, the Bland-Altman plot describes the important relationship, that, with increasing mean aJPS error, there exists an increase in variability of the aJPS error measurements (Fig 7).

## Discussion

In this study, three separate datasets demonstrated that good within-session reliability can be achieved with active joint position sense error (aJPS) using app-based digital inclinometry.

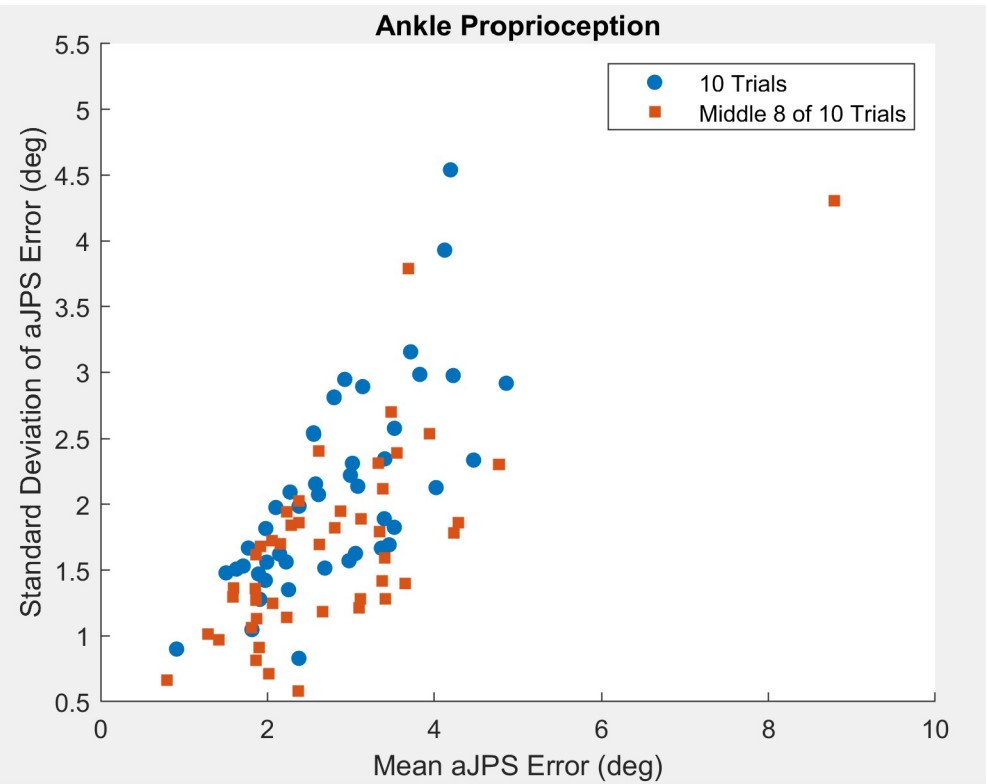

**Fig 6. Scatter plot comparing ankle aJPS error standard deviation and mean aJPS values measured by the digital inclinometer application.**

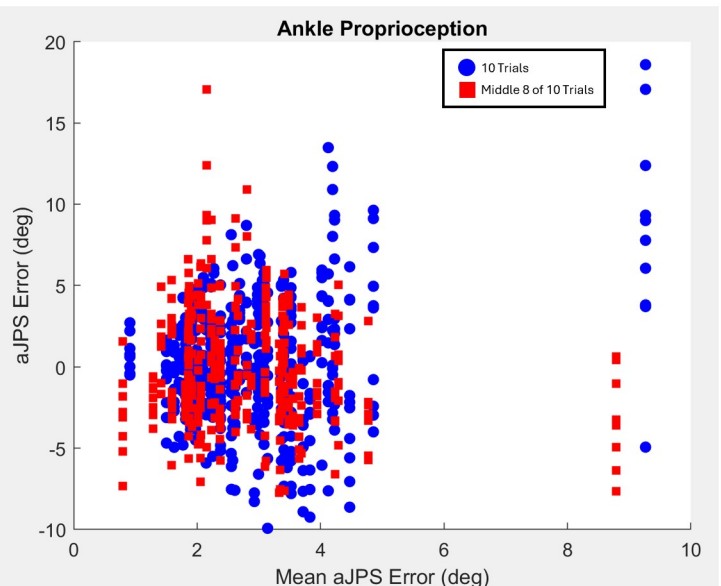

**Fig 7. Sample Bland-Altman plot comparing raw ankle aJPS error values and the absolute mean aJPS error values measured by the digital inclinometer application.**

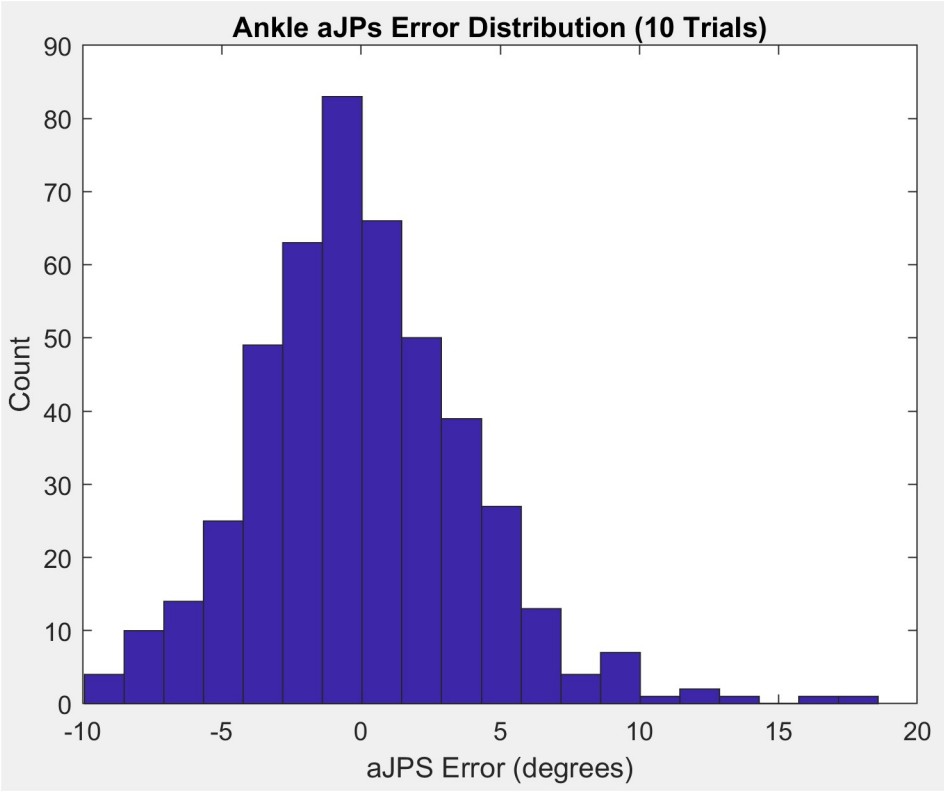

**Fig 8. Sample histogram plot of the distribution of raw ankle aJPS error values measured by the digital inclinometer application.**

Both the hip and knee joints achieved good reliability with only five trials, where the ankle joint required up to ten trials to achieve similar reliability results. This could be due, in part, by the ankle's position being more distal than the other two joints studied, making the measurement of ankle proprioception more variable by nature.

This way of assessing proprioception reduces the clinical barriers mentioned in this article, such as expensive, bulky equipment and inefficient and complicated data analysis. Using a digital inclinometry app, as used in the current study, is simple to use, inexpensive, and easily portable. These are all qualities clinicians look for in a clinical tool, meaning this could be easily implemented in the clinical setting as a means of gaining valuable quantitative data on patient proprioception.

## Hip proprioception

Results of the hip proprioception testing showed good test-retest reliability of the digital inclinometer app when recording five repeated trials. Results of the hip proprioception testing, analyzed between limbs, showed that there was moderate correlation in the joint position sense data. This suggests that one cannot assume that healthy individuals are symmetric in their hip proprioception.

These results build upon the results of a previous study, aiming to assess the validity and intra-rater reliability of a smartphone application called "Clinometer" for measuring hip, knee, and ankle sagittal ranges of motion, using a digital inclinometer as the reference standard [26].

In this study, Mohammad et al. found that the "Clinometer" app displayed excellent validity when compared to the digital inclinometer for hip movements. Additionally, the "Clinometer" app demonstrated excellent reliability for hip sagittal plane motion. Given that both Mohammad et al.'s assessment of range of motion and our measurement of joint position sense error require accurate measurements of segment positions relative to gravity, the current study's results support and build upon the results of this previous study by showing comparable reliability values, good reliability for joint position sense error at the hip, and moderate reliability when comparing between limbs.

### Knee proprioception

Results of the knee proprioception testing showed good test-retest reliability of the digital inclinometer app when recording five repeated trials. The knee proprioception pre and post an intervening activity demonstrated poor correlation measurements separated by a delay in time, leading to some hesitation in trusting that one's aJPS remains the same from visit to visit. However, it should be noted that in this knee dataset, in between tests participants performed several activities including single-leg landings and isokinetic strength testing. Therefore, we do not know at present whether the differences seen between knee aJPS sessions were due to natural variation over time or due to the intervening activities altering one's JPS. A recent study looking into effects on proprioception immediately following exercise showed a decrease in proprioceptive ability directly after completion of the exercise and returned to pre-exercise levels after 72 hours of passive recovery [27]. This may suggest that proprioception can be affected due to intervening activities and exercises, supporting our results of poor correlation of knee proprioception before and after the activities.

The knee proprioception reliability results are supported by other literature, such as a study performed to assess the reliability of a custom-built web application running on a smartphone, with an isokinetic dynamometer used as the main control [28]. In this study, Ravi concluded that the standard deviations obtained were less than 0.5 degrees for the knee angle measurements on the web-application against the isokinetic dynamometer. The statistical analyses demonstrated no significant difference between the web application and the isokinetic dynamometer. These data together demonstrate the high degree of reliability and validity of the web-based measurement of knee joint angle, closely aligned to measurements obtained on the isokinetic dynamometer [28]. The current study builds upon these results by showing comparable reliability values when measuring knee joint position sense error, as well as showing little difference between data collected before and after an intervening activity.

### Ankle proprioception

Results of the ankle aJPS testing showed poor test-retest reliability of the digital inclinometer app when recording only five repeated trials, but good reliability when 10 trials were collected. This result suggests that ankle proprioception is more variable than the knee and hip, but future research is necessary to test this hypothesis. Care should be taken in the future to ensure enough trials are collected for a reliable ankle proprioception assessment.

These results build upon data collected and analyzed in a prior study that aimed to test the reliability of smartphone goniometry in the ankle joint by comparing it with a universal goniometer and to assess inter-rater and intra-rater reliability for the smartphone goniometer record application [29]. In this study, Alawna et al. found that, for measuring ankle dorsiflexion range of motion (ROM), both instruments showed excellent interrater reliability. Intra-rater reliability was excellent in both instruments in ankle dorsiflexion, and as for measuring ankle plantarflexion, both instruments showed excellent interrater and intra-rater reliability.

Similarly to the hip proprioception literature previously mentioned, these data show great reliability when comparing smartphone and universal goniometry joint angle measurements. The current study extends these results by comparing ankle joint position sense error values and by determining the number of trials that must be collected to establish good reliability.

## Limitations

It is important to highlight that the data described in this paper are made up of three separately collected datasets. Each dataset, for the hip, knee, and ankle aJPS, was individually collected at different times, by different testers, in different environments, and with different participants. In the future, it may be more appropriate to collect a single dataset with the same sets of testers and participants, allowing for more direct comparison of the joint measurements to one another.

Some important considerations when using digital inclinometer apps in the testing protocols described above, or similar, include: ensuring the sensor is attached in the correct orientation; ensuring the sensor is attached securely to each participant, so as to minimize any extra noise or error from it shifting its position during dynamic tasks; and ensuring that whatever "learned position" remains consistent across participants and trials, whether that be through the use of a device or environmental cue. In addition, a specific digital inclinometer app was used for this study, and no comparison was made to other available digital inclinometers. This limitation is considered to be minor, because both the inertial sensors and data processing to estimate device angle represent common standards across the many available digital inclinometers such as the EasyAngle® digital goniometer (Meloq AB, Stockholm, Sweden) or the RST Instruments digital inclinometer (RST Instruments Ltd, Maple Ridge, Canada).

Future research should include a more robust device to cue the "learned position", taking care to ensure that the design remains fairly simple to translate to a clinical setting, so that the participant does not change position with respect to the apparatus throughout the trials for better and more consistent data collection. In addition, more diversity of the population in age, joint pathologies or neurological conditions and a higher number of participants would improve the generalizability of the results.

## Conclusion

A digital inclinometer app achieved good intra-session reliability at measuring joint position sense in the hip, knee, and ankle with an appropriate number of trials and a trimmed-average approach. These results should be generalizable across the many devices on the market that can be used similarly to the digital inclinometer app system in this study. This tool and methodology can be considered for use in both clinical and research environments to measure proprioception and proprioceptive deficits.

## Supporting information

**S1 Raw data.**
(XLSX)

## Acknowledgments

The authors would like to thank Al Caserta, Pierce Fonseca, Atticus Jordan, Nelson Glover, Andrea Wanamaker, Kenechukwu Okoye, Michael McNally, Kimberly Bigelow, Paige Ingram, Sarah Hollis, Amanda Delaney, Jack Schultz, Jennifer Perry, Chelsea Jentsch, Cruz Finnicum, Cody Ballay, and Orion Swanson for their time and hard work involved in the collection of the

data reported in this paper. We would also like to thank the American Society of Biomechanics for allowing and supporting our data collection during the 2017 annual meeting in Boulder, Colorado.

## Author Contributions

**Conceptualization:** Rachel H. Teater, Megan M. Bals, Scott M. Monfort, Ajit M. W. Chaudhari.

**Data curation:** Sophia G. Chirumbole, Rachel H. Teater, Daniel R. Richie, Ajit M. W. Chaudhari.

**Formal analysis:** Sophia G. Chirumbole, Rachel H. Teater, Megan M. Bals, Scott M. Monfort, Ajit M. W. Chaudhari.

**Investigation:** Sophia G. Chirumbole, Megan M. Bals, Scott M. Monfort, Ajit M. W. Chaudhari.

**Methodology:** Megan M. Bals, Scott M. Monfort.

**Supervision:** Ajit M. W. Chaudhari.

**Visualization:** Sophia G. Chirumbole.

**Writing – original draft:** Sophia G. Chirumbole, Rachel H. Teater, Scott M. Monfort, Ajit M. W. Chaudhari.

**Writing – review & editing:** Sophia G. Chirumbole, Rachel H. Teater, Megan M. Bals, Daniel R. Richie, Scott M. Monfort, Ajit M. W. Chaudhari.

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
