## [Decision Letter · Decision Letter 0]

16 Apr 2024

PONE-D-24-02242Assessing the Reliability of a Digital Inclinometer App for Measuring Hip, Knee, and Ankle ProprioceptionPLOS ONE

Dear Dr. Chaudhari,

Thank you for submitting your manuscript to PLOS ONE. After careful consideration, we feel that it has merit but does not fully meet PLOS ONE’s publication criteria as it currently stands. Therefore, we invite you to submit a revised version of the manuscript that addresses the points raised during the review process.

Please submit your revised manuscript by May 31 2024 11:59PM. If you will need more time than this to complete your revisions, please reply to this message or contact the journal office at plosone@plos.org. Please include the following items when submitting your revised manuscript:A rebuttal letter that responds to each point raised by the academic editor and reviewer(s). You should upload this letter as a separate file labeled 'Response to Reviewers'.A marked-up copy of your manuscript that highlights changes made to the original version. You should upload this as a separate file labeled 'Revised Manuscript with Track Changes'.An unmarked version of your revised paper without tracked changes. You should upload this as a separate file labeled 'Manuscript'.

We look forward to receiving your revised manuscript.

Kind regards,

Ravi Shankar Yerragonda Reddy, Ph.D

Academic Editor

PLOS ONE

Journal Requirements:

Reviewers' comments:

Reviewer's Responses to Questions

**Comments to the Author**

1. Is the manuscript technically sound, and do the data support the conclusions?

Reviewer #1: No

Reviewer #2: Yes

Reviewer #3: Yes

2. Has the statistical analysis been performed appropriately and rigorously? 

Reviewer #1: No

Reviewer #2: Yes

Reviewer #3: Yes

3. Have the authors made all data underlying the findings in their manuscript fully available?

Reviewer #1: No

Reviewer #2: Yes

Reviewer #3: Yes

4. Is the manuscript presented in an intelligible fashion and written in standard English?

Reviewer #1: Yes

Reviewer #2: Yes

Reviewer #3: Yes

5. Review Comments to the Author

**Reviewer #1:** Abstract: It is recommended to include the 95% confidence intervals (CI) for the ICC values.

Introduction: No comments

Methods

• To strengthen the methodology, consider providing more details on how the sample size was calculated. This will enhance the transparency and generalizability of the study.

• It would be helpful to explain the rationale behind choosing ICC (2,1) and ICC (2,k) as the specific intraclass correlation coefficients. Knowing the justification for these selections will improve the clarity of the research design.

• Statistical analyses: Including standard error of measurement data would be beneficial. This additional information will provide a more comprehensive understanding of the measurement reliability.

Results

• The results section would benefit from reporting the mean difference between the readings along with their corresponding significance levels. This will allow readers to better interpret the statistical significance of the findings.

• To improve the clarity and precision of the findings, it's recommended to include the 95% CI for the ICC values.

• Presenting the data in a well-organized table would enhance readability and facilitate easier comparison between different groups or variables.

• I recommend using Bland-Altman plots to visualize potential systematic bias between the readings.

Discussion & Conclusions: No comments

**Reviewer #2:** Dear author,

Kindly specify the number of trials you have undertaken for each joint specifically. Rest everything is written precisely. The aims and objectives are specified correctly and are in correspondence with the research question, need of the study, and conclusion.

Thank you

**Reviewer #3: **Dear Authors

Thanks for your interesting article. I really enjoy it. However, in my opinion,some issues should be addressed.

1. I think you could elaborate the statistical analysis for better results. Why did not you use SEM, MDC or even Cronbach's alpha? It seems the possible error of measurement should be calculated to make sure that the measurements are not because of errors.

2. For using new apps or even new devices, finding validity of the device is necessary. However, How did you know the app is valid? have you checked it with a gold standard?

6. PLOS authors have the option to publish the peer review history of their article (what does this mean?). If published, this will include your full peer review and any attached files.

Reviewer #1: No

Reviewer #2: **Yes: **Sakshi P. Arora

Reviewer #3: No

---

## [Author Response · Author response to Decision Letter 0]

26 Jun 2024

Please see the attached Response to Reviewers file, which contains formatting, updated tables and figures.

---

## [Decision Letter · Decision Letter 1]

30 Jul 2024

Assessing the Reliability of a Digital Inclinometer App for Measuring Hip, Knee, and Ankle Proprioception

PONE-D-24-02242R1

Dear Dr. Ajit M.W. Chaudhari,

We’re pleased to inform you that your manuscript has been judged scientifically suitable for publication and will be formally accepted for publication once it meets all outstanding technical requirements.

Kind regards,

Ravi Shankar Yerragonda Reddy, Ph.D

Academic Editor

PLOS ONE

Additional Editor Comments (optional):

I am pleased to inform you that your paper has been accepted for publication. The reviewers have recommended your publication, recognizing its significant contribution to the field. Congratulations on this achievement.

Reviewers' comments:

Reviewer's Responses to Questions

**Comments to the Author**

1. If the authors have adequately addressed your comments raised in a previous round of review and you feel that this manuscript is now acceptable for publication, you may indicate that here to bypass the “Comments to the Author” section, enter your conflict of interest statement in the “Confidential to Editor” section, and submit your "Accept" recommendation.

Reviewer #1: All comments have been addressed

Reviewer #3: All comments have been addressed

2. Is the manuscript technically sound, and do the data support the conclusions?

Reviewer #1: Yes

Reviewer #3: Yes

3. Has the statistical analysis been performed appropriately and rigorously? 

Reviewer #1: I Don't Know

Reviewer #3: I Don't Know

4. Have the authors made all data underlying the findings in their manuscript fully available?

Reviewer #1: Yes

Reviewer #3: Yes

5. Is the manuscript presented in an intelligible fashion and written in standard English?

Reviewer #1: Yes

Reviewer #3: Yes

6. Review Comments to the Author

Reviewer #1: I have no further comments for the authors. Thank you.

Reviewer #3: Thanks for inserting table 1, however, if it is possible please mention SEM instead od standard error in the last column of table 1.

Well done.

7. PLOS authors have the option to publish the peer review history of their article (what does this mean?). If published, this will include your full peer review and any attached files.

Reviewer #1: No

Reviewer #3: No

---

## [Editor Report · Acceptance letter]

5 Aug 2024

PONE-D-24-02242R1 

PLOS ONE

Dear Dr. Chaudhari, 

I'm pleased to inform you that your manuscript has been deemed suitable for publication in PLOS ONE. Congratulations! Your manuscript is now being handed over to our production team.

Kind regards, 

on behalf of

Dr. Ravi Shankar Yerragonda Reddy 

Academic Editor

PLOS ONE